# Beyond the Phenomenology of the Inconspicuous †

Carla Canullo

Department of "Studi Umanistici", Philosophy, University of Macerata, 62100 Macerata, Italy;
carla.canullo@unimc.it

† This text was translated from the Italian by Bruno Cassarà (Fordham University), whom I thank for his engagement with the text and for his valuable comments.

**Abstract:** How does spirit appear? In fact, it does not appear, and for this reason, we could refer to it, following Heidegger, as "inconspicuous" (*unscheinbar*). The Heideggerian path investigates this inconspicuous starting from the Husserlian method, and yet, this is not the only Phenomenology of the "Inconspicuous" Spirit: Hegel had already thematized it in 1807. It is thus possible to identify at least two Phenomenologies of the "Inconspicuous" spirit. These two phenomenologies, however, do not simply put forth distinct phenomenological methods, nor do they merely propose differing modes of spirit's manifestation. In each of these phenomenologies, rather, what we call "spirit" manifests different traits: in one instance, it appears as absolute knowing, and, in the other, it manifests "from itself" as "phenomenon". Yet how, exactly, does spirit manifest "starting from itself as phenomenon"? Certainly not in the mode of entities, but rather in the modality that historical phenomenology, which also includes Edmund Husserl's work, has grasped. A question remains, however: is the inconspicuous coextensive with "spirit"? Certainly, spirit is inconspicuous, but it is not only spirit that is such. A certain phenomenological practice understood this well, a practice that several French authors have pushed. Jean-Luc Marion, Michel Henry, and Jean-Louis Chrétien have all contributed, in a certain way, to the phenomenology of the inconspicuous. However, do these authors carry out a phenomenology of inconspicuous *spirit*? Perhaps what French phenomenology gives us today, after an itinerary that has discovered several senses of the inconspicuous, is precisely the return to spirit that is missing in, and was missed by, this tradition.

**Keywords:** phenomenology; inconspicuous; spirit; Hegel; Marion; Henry; Chrétien

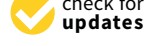



## 1. Introduction

Spirit does not appear as does any other being or thing, and yet it appears. It is precisely in virtue of this peculiar manifestation that a person may be said to be "spiritual", a term that normally refers to someone who leads a life that privileges its "interior" dimension over carnality and materiality. Perhaps this is what one usually means by "spirit": what is not material, what is ethereal, volatile, in-visible. It is spoken of, but it is not manifest as a visible thing. Nevertheless, it appears, for if it were not to appear, one would be unable even to speak of it. It appears (and is spoken of) without manifesting as object or thing. For this reason, it is inconspicuous and appears *as* inconspicuous. However, what does it mean to appear *as* inconspicuous? To answer the question, we will proceed in this way: 1—we will start from the meaning of the "Phenomenology of the Inconspicuous" according to Heidegger and 2—from the meaning of phenomenology as a tautology that the German philosopher proposes both when he debates about "Phenomenology of the Inconspicuous" and (in our opinion) in his reading of Hegel's *Phenomenology of the Spirit*. 3—Thereafter, we will see if and in what way Heidegger, after this reference to Hegel, spoke of the Spirit. In this regard, the contribution of Derrida, reader of Heidegger and of *Unterwegs zur Sprache*, will be the main reference. 4—Then, we will see if the philosophers who according to Dominique Janicaud were inspired by Heidegger's "Phenomenology of the Inconspicuous" (i.e., Michel Henry, Jean-Luc Marion, and Jean-Louis Chrétien) have

also maintained the reference to the inconspicuous Spirit or not and, if not, 5—we will see what, in our opinion, a reference not marginal but central to the Spirit brings to the thought of the three French philosophers.

## 2. Which Phenomenology of the Inconspicuous?

Heidegger takes up the inconspicuous in his Zähringen Seminar of 1973 when, in speaking of Being (*Sein*), he states that the Greek *einai* means "to presence". Thus, Heidegger translates the Parmenidean formula "esti gar einai" as "presencing namely presences" (*anwest nämlich Anwesen*).[1] This assertion, he continues, is a tautology because "it names the Same only once, and indeed as itself" (Heidegger 2003, pp. 134–36). Here, then, is the sphere of the inconspicuous: it "presences presencing itself", but this is a tautology. Now, Heidegger concludes, the sense of phenomenology is precisely this tautology, and the original sense of phenomenology is its essential character as a tautological form of thinking.

That phenomenology is tautological means that this way of thinking does nothing other than make manifest the Same, so that what is at the origin, that for which everything is and of which everything consists, is precisely what is found at the end. Heidegger himself makes this point, stating that "phenomenology is a path that leads away to come before . . . , and it lets that before which it is led show itself" (Heidegger 2003, pp. 136–37). Phenomenology, or better yet, phenomenology thus understood, is a phenomenology of the inconspicuous. The difficulty of these few lines is not lost on us: what does it mean to say that phenomenology is a tautological thinking? What does it mean that the inconspicuous is what is arrived at? For Heidegger, phenomenology is the *path* that leads us toward Being, and Being is our home. The German philosopher explicates this when he writes of the origin of thinking: being at home where thinking originates itself means "to attain a grounded residence in Dasein where thinking receives the determination of its essence" (Heidegger 2003, p. 93). As a path, phenomenology leads toward that which does not appear as things or beings (*Seiende*), to where thinking dwells, that is, toward Being. This tautology, however, is not new to Heidegger's thought. This is confirmed by certain passages in the "Preliminary Considerations" to his reading of Hegel's *Phenomenology of Spirit*, a work to which Heidegger dedicated several courses in Freiburg between 1930 and 1931.

"The end is the beginning which has only become other and thus come to itself" (Heidegger 1988, p. 36). This is how Heidegger reads Hegel's *Phenomenology of Spirit*, and he further explains that an understanding of the end is *plainly* indispensable in the case of Hegel and for understanding his basic intention and inquiry, which begins and must begin with absolute knowledge. This is so because the end is already plainly the beginning and because the *way in which* the end is the beginning (and vice versa) has already been decided (see Heidegger 1988, p. 37). Moreover, he concludes, "The meaning of this way of being is determined precisely from and with absolute knowledge itself" (Heidegger 1988, p. 37).

In the same passage, Heidegger states that "the *Phenomenology of Spirit* begins absolutely with the absolute" (Heidegger 1988, p. 37), and the absolute is Spirit. Now, is this absolute beginning with the absolute not itself a tautology? Furthermore, the fact that the end is the beginning and vice versa—is this not also a tautology? It is difficult to say otherwise. In this reading of the *Phenomenology of Spirit*, the tautology of Spirit from which one must begin responds to the tautological inconspicuous, the inconspicuous as what is arrived at. Is Spirit, however, inconspicuous? What is the relationship between the inconspicuous *toward which* one is and Spirit as the *beginning* which has always been? It is difficult to give an answer because Heidegger himself, who only left few indications on the inconspicuous and on Spirit, never juxtaposed the two. *Is* Spirit, however, inconspicuous?

It is certainly not inconspicuous in the way Heidegger means this term. However, what does Heidegger mean by a "Phenomenology of the Inconspicuous?" An article by Jason W. Alvis, "Making Sense of Heidegger's 'Phenomenology of the Inconspicuous' or Inapparent (*Phänomenologie des Unscheinbaren*)" (Alvis 2018, pp. 211–38), undertakes a detailed examination both of Heidegger's texts and of the importance of such a phe-

nomenology. Alvis proposes three possible interpretations. The first "is the most likely to be accepted by readers of Heidegger due to its generality: *Unscheinbarkeit* is a character trait that any and all phenomena are capable of enacting, and therefore inherent to a Heideggerian phenomenology" (Alvis 2018, p. 230). The second posits "a new, particular step [ . . . ] introduced into all of phenomenology that involves ones [sic] turning attention to the various modes of potential hiddenness (*Verborgenheit* and its cognates) within all phenomena, and "inconspicuousness" is now to be included as a form, mode, or "manifestation" among them. In which case, one can only get at these particularly "dark" or obscure corners of experience within the intuition through actually performing the "exercises" of such a phenomenology" (Alvis 2018, p. 233).

The third interpretation intends the inconspicuous as "a direct reference to specific, unique, and distinct phenomena that paradoxically exceed the visible/invisible (*sichtbar/unsichtbar*) polarity, yet still somehow are present and affective" (Alvis 2018, p. 233). There are numerous other insights that deserve to be pointed out, but I will limit myself to quoting Alvis' conclusion in full:

> "Ultimately, a phenomenology of the inconspicuous should matter to us because it aims to take seriously what the sharp distinctions between appearance/non-appearance have generally sought to veil through their being cast in various matrixes of opposition. How we take it that certain things are no longer worth our attention, how we take people and family for granted, and how we overlook the least of those around us by merit of 'shinier' things, celebrities, or spectacles that command our attention, all provide some reasoning for why this topic is an important one. Overwhelming privilege is accorded today to whatever can present the greatest possible degree of unconcealment: the greater degree the spectacle, we are complicit to believe, the more sacred an event becomes and the closer to divinity it presents itself. This bears consequences for what has become most familiar, and therefore insignificant to us. Yet with a bit of optimism, phenomenology may provide tools for sharpening our ability to take seriously again what is ordinary and familiar". (Alvis 2018, pp. 236–37)

Alvis is right. On one hand, we cannot expect to define with precision what Heidegger only alluded to, while on the other, not to define it is not the same as ignoring its importance or the contribution it can make to our way of looking at reality. Yet, in highlighting the importance of the inconspicuous, have we lost sight of Spirit? Yes, if we follow the phenomenology of the inconspicuous, since Heidegger never discusses Spirit there. No, if we return to the beginning of this section after having followed Heidegger in his reading of Hegel and after having found in Hegel's *Phenomenology* the same tautology as that of the Phenomenology of the Inconspicuous. At this point we must ask, what is a tautology and, more importantly, how does it appear?

## 3. Regarding Tautology, or Spirit Is Life

In classical formal logic, tautology means that a proposition simply repeats in its predicate what is already said in its subject, although the proposition purports to be defining something. In simpler terms, a proposition is true because it is true—this is what tautology means in logic. Can we say that the phenomenology of the inconspicuous is tautological in this sense? Probably not, and this is for two reasons. First, the expression that was said to be tautological, "presencing namely presences", does not simply repeat the same but brings a surplus: that Being is present means that when Being is what presences, Being is not some present thing but rather an event, a self-manifestation, a self-presenting. Thus, the expression is not tautological because in saying that Being is present, it also says that what is present is also what presents, that is, what is present is what enacts the movement of self-presentation. Actually, movement is nothing; it is an act that brings something before the eyes, but which in itself is nothing. Therefore, movement is not "present", but it is what allows the present thing "to present itself", to "come out" and let

itself be seen. We, however, never see this movement but only the one who (or that which) is present. Hegelian Spirit should also be understood this way.

Heidegger is right when he writes, as already quoted above, that in the *Phenomenology of Spirit* "the end is the beginning which has only become other and thus come to itself", and that "the *Phenomenology of Spirit* begins absolutely with the absolute". Nevertheless, there are at least two questions we could ask, that is: (1) What does this mean? and (2) how, exactly, does this happen? The answer to (1) is simple: it is a tautology, yes, but its repetition is of a kind that unfolds and, in unfolding, makes manifest something *new*. Why does this happen? Put differently, how does this happen exactly? It happens because Spirit is Life.

As Hegel explains in his *Phenomenology of Spirit* and at the end of his *Science of Logic*,[2] Spirit is Life, and Life is at once the individual living thing, the vital process by which all living things are such, and the "life of the Concept" and of thought. Life is the *vital* unity of these three moments and the presupposition of the knowing that grasps it. It is in fact knowing and "the necessity proper to the Concept" that introduce the idea of "the true in and for itself", and this idea is first and foremost Life. However, Spirit is also *absolute* along with being Life, and for this reason, includes and annexes all that is part of the historical journey of finitude, confirming in this way the Heideggerian comment that "the *Phenomenology of Spirit* begins absolutely with the absolute". The absolute, however, is Life, and as such unfolds in and through the multiplicity of living things. Furthermore, Life as such is inconspicuous, it never appears as itself but makes manifest the living *as* living, at once self-manifesting and making manifest each living thing. In fact, living things appear as such through and because of Life.

In Hegel, therefore, living things appear through and because of Life which, in turn, is Spirit. The latter, then, is grasped in the Concept or, more precisely, in the Idea that expresses its manifestation in and for itself in the representation of it that Knowing grasps. Thus, the Concept as Idea is the way that Spirit grasps itself and, therefore, the manifestation of Life is possible for Absolute Knowing. In this way, Spirit is a tautology that, in its own happening, makes to happen something other, namely, Life.

### 4. Phenomenologies without Spirit?

If we were to ask how Spirit unfolds, the Hegelian answer would have to be that Spirit unfolds as Concept and Idea, thus as *representation*, and that after Hegel's *Phenomenology*, there have been no further phenomenologies of Spirit. Is a phenomenology of Spirit as "Life", as we find in Hegel, possible without Spirit manifesting itself through the representation? Not if we remain with Hegel, but perhaps it should be possible if we turn to other phenomenologies. However, in these other phenomenologies, the absence, or near-absence, of Spirit is glaring.

This near-absence is perhaps more descriptive of post-Husserlian phenomenology and less of Edmund Husserl himself. It is especially true if we consider, among many others of his writings, the Freiburg lecture courses by the title of *Natur und Geist* taught in 1919 and 1927, the latter of which must be read alongside his celebrated *Crisis of the European Sciences*. In the 1927 iteration of the course, Husserl writes that "the philosophical problems of Nature and Spirit do not lie 'next to' those of the natural and spiritual sciences (*geisteswissenschaftlichen*), as if separate from them",[3] and insists a few pages later that it is impossible to fully separate Nature and Spirit (see Husserl 2001, p. 16). Here, Husserl is reckoning with the philosophies of Dilthey, Windelband, and Rickert, and starting from these, he confronts the "Nature/Spirit" question. This question, however, does not posit Spirit as such as the central theme of phenomenology. Spirit makes an appearance but is not at the heart of phenomenological unfolding. On the other hand, Spirit is almost absent from the French phenomenology that Dominique Janicaud included in the "Theological Turn" (or at least it is not the core of these phenomenologies, even if it is present), a philosophical tradition that did not merely face, but took up and took over Heidegger's Phenomenology of the Inconspicuous. According to what we have seen in Heidegger, such an absence is

not surprising at all. We might even ask if these are "phenomenologies without Spirit" precisely because of their Heideggerian inheritance. Janicaud understood well the weight of this inheritance: "Without Heidegger's *Kehre*, there would be no theological turn" (Janicaud 2001, p. 31), because already in Heidegger himself it is necessary to interpret the Phenomenology of the Inconspicuous as "[permitting] the retrieval of both 'givenness' and the most originary dimension of temporality to attain and bind together the traces of a new approach [to] the Sacred" (Janicaud 2001, p. 31). A confirmation of Janicaud's conclusions would require a return to the original sense of the Phenomenology of the Inconspicuous. However, as we have seen, it is impossible to give a univocal interpretation of this phenomenology, as Alvis shows, and this essential ambiguity would in turn require an analysis of the individual cases in which these conclusions are revealed to be true. At the same time, it is indeed true that some phenomenologists in France have opened ways to think an inconspicuous Life (Michel Henry), an inconspicuous flesh (Maurice Merleau-Ponty, Michel Henry), givenness (Jean-Luc Marion), and the event (Jean-Luc Marion, Claude Romano). These are paths followed in order to think the appearing of "things" outside of horizons that would regulate the possibility of such appearing (for instance, the transcendental subject/Ego or Being). These are phenomenologies that seek the condition of possibility of phenomena in an "inconspicuous" that is nevertheless efficacious. This inconspicuous would appear as it brings to manifestation phenomena that would not be experienceable within horizons such as the world, Being, or subjectivity. Furthermore, this inconspicuous would manifest in phenomena that would not be phenomena at all if inserted into the traditional conditions that phenomenology has already investigated. These are phenomena such as Life, which, as we have seen, never manifests as such. For instance, since Life lacks an adequate "phenomenology of Life" that would seek the conditions of its appearing, this phenomenon is left to inquiries other than phenomenology, that is, to the so-called *Geisteswissenschaften*.

This is probably the reason for which these authors did not make the Spirit the main core of their philosophy, with the exception of Michel Henry when he discussed spirituality in *Barbarism* (Henry 2012). Therefore, we are faced with an apparent contradiction: even if, generally speaking, it is only partially correct to maintain that the work of some French phenomenologists is the direct consequence of the Phenomenology of the Inconspicuous (in the first place, because it is impossible to speak of *a single* Phenomenology of the Inconspicuous), the absence of Spirit in fact brings these authors closer to Heidegger, in whose works we find the same lack of reflection on this theme. Why is Spirit absent? Perhaps because *Geist* is a horizon which, due to its Hegelian heritage, belongs to the realm of conceptuality, and this would impose ahead of time the manner in which Spirit can appear. By contrast, Heidegger intends the phenomenon as what manifests starting not from Spirit but from itself—a definition which the phenomenologists of the so-called "Theological Turn" have never put into question and have accepted as a secure point of departure. To think of a phenomenon in the light of Spirit, then, would impose the latter as a condition of possibility for its appearing, yet another condition along with the transcendental Ego (Husserl) and Being or world (Heidegger).

Yet is this the only possible conclusion? Could Spirit not be the name for the originary character of appearing itself rather than a condition of possibility or a new phenomenological horizon? In *Of Spirit: Heidegger and the Question*, Jacques Derrida opens the way for an affirmative answer to this question. Derrida reminds the reader that Heidegger mentions Spirit as he comments on Georg Trakl's last poem in "Language in the Poem: A Discussion on Georg Trakl's Poetic Work" (Heidegger 1971). In those pages, Heidegger writes that the human being is spiritual in that "the nature of spirit consists in a bursting into flame, it strikes a new course, lights it, and sets man on the way" (Heidegger 1971, pp. 179–80). Spirit sets on the way, it moves and displaces. It is *because* and *while* it acts and moves. Toward where? Perhaps to that "grounded residence in Dasein where thinking receives the determination of its essence", toward Being which, however, is *inconspicuous*. Commenting on this Heideggerian Spirit, Derrida writes that it precedes both *pneuma* and *spiritus*, that it



is neither of these but rather it is something more originary, the possibility itself that Spirit would give itself, and also that "It opens onto what remains *origin-heterogeneous*".[4] In this sense, writes Derrida, the Heideggerian Spirit is *spark/Frühe*, and "it is because *Geist* is flame that there is *pneuma* and *spiritus*" (Derrida 1989, p. 97).

Thus, Spirit can be the originariness of all appearing, and Derrida stresses this precisely as he reads Heidegger, bringing to light a sense of Spirit different from that of Hegel. It is necessary, however, to find phenomena that confirm this, and in order to do so, I shall enact a double departure and a double overcoming. It will be necessary to overcome the Phenomenology of the Inconspicuous which, on its part, does not take up Spirit, and it will also be necessary to overcome Hegel, who, though he conceives of Spirit as a phenomenon that appears starting from itself, nevertheless understands it through the Concept and therefore within the framework of representation. Overcoming Hegel does not mean, however, abandoning the irrevocable legacy that he leaves behind, that is, the discovery that Spirit manifests as Life. If, for Hegel, Life is Spirit as an Idea that lets itself be comprehended, we will have to see whether Life itself can be apprehended as a phenomenon that appears starting from itself. This has been done, at least from a certain point of view, by Michel Henry and, from the point of view of the call and especially the call to life, by Jean-Luc Marion and Jean-Louis Chrétien. I will not be summarizing the entire thought of these authors, all the more because they are already known, and will instead limit myself to an attempt at answering the following question: what does the manifestation of Spirit add to these phenomenologies?

## 5. Spirit, the Origin of Life

Let us retrace our steps. Is it possible for Spirit to appear without recourse to Being or to the Idea, outside of the Concept and of representation? Yes, if we take up an aspect that Derrida highlights in his reading of Heidegger, namely, that Spirit is the initial spark (*Frühe*) that precedes *pneuma* and *spiritus*. Spirit is a spark, what enflames and brings to life, what warms and emanates energy. However, it is also breath that enlivens and ensouls. Thanks to Spirit, in fact, flesh becomes animate, while without Spirit it is dead. What phenomena appear through the event of this spark or breath that animates and gives life? To avoid misunderstandings, Henry, Marion, and Chrétien will be discussed separately. I will show how with each of them, a new, non-Hegelian phenomenology of Spirit leads the appearing of phenomena back to their original source. Furthermore, I will show what Spirit adds to their phenomenologies. Spirit would be, in my view, what precedes and originates all differentiation and, as such, is the originary source that gives itself as tautology—in the sense of what reiterates itself while giving what is other than itself.

### 5.1. Spirit and Praxis (Michel Henry)

Henry's phenomenology is widely known as a phenomenology of Life, not of Spirit. Of course, one could always respond that any phenomenon of Life is a manifestation of Spirit, since Spirit and Life, as we saw with Hegel, are inextricably tied. Furthermore, the characteristic that sets apart Hegelian Spirit and the Henry's Life is the same, namely, *praxis*. If, in Hegel's *Phenomenology*, Spirit appears as action and action, more specifically, as a *praxis* that is able to shape things through labor,[5] then for Henry, Life itself is *praxis*.

We know that *praxis* is one of the Aristotelian virtues the end of which is internal to itself.[6] *Praxis* is productive similar to *poiesis*, but the end of the former is not some external product such as an artifact—the end is *acting itself*. However, what is the essence of acting? An Aristotelian would have no precise answer here because *praxis* is an *ethos*, a *habitus* that belongs to the human being so intimately that it is nonsense to ask about its essence. Henry, on the other hand, gives an answer to this question in the works that discuss *praxis* (*Marx* (Henry 1983) and *Barbarism*), as well as in *I am the Truth* (Henry 2002), the first work in which he deals with Christianity. This text begins with an inquiry into the nature of truth, which Henry intends as what explains the difference between those phenomena that appear in the world and phenomena which cannot appear in the world.

Life belongs among the latter. Henry writes about Life as a phenomenologist and from a phenomenological point of view, opposing world and Life and their truths. "What is true," Henry writes, "is what shows itself . . . . It is this appearance as such . . . that constitutes the 'truth'"(Henry 2002, p. 12). As truth consists of "the pure fact of showing itself, or else of appearing, of manifesting itself, of revealing itself, we can just as well call truth 'monstration', 'apparition', 'manifestation', 'revelation.'"(Henry 2002, p. 12).

Henry's thesis is thus that truth would consist in the *pure act of self-showing* involved in all that appears, and the essence of truth would be the pure and simple fact of self-showing: "It is only because the pure act of appearing takes place, and that, in it, the truth deploys its essence beforehand, that everything that appears is susceptible of doing so . . . . Thus, any truth concerning thing—beings [*étants*], as the Greeks said—any ontic truth, refers back to a pure phenomenological truth that it presupposes, refers back to the pure act of self-showing, considered in itself and as such" (Henry 2002, p. 12). From this thesis will be born Henry's well-known separation of world and Life, or better put, of *the truth of* the world and *the truth of* Life. The former then develops in itself another separation between truth and the true because it consists of a centrifugal motion, that is, of a movement of "pushing out" and exposition. Phenomena are *true* to the extent that they are exposed within the horizon of the world. Otherwise stated, the world is the background starting from which "objects" are phenomenalized: "A thing exists for us only if it shows itself to us as a phenomenon. And it shows itself to us only in that primordial "outsideness" that is the world. It matters little in the end whether the truth of the world is understood through consciousness or through the world itself, if in either case what constitutes the capacity of self-showing, truth, manifestation, is "ousideness" as such" (Henry 2002, pp. 15–16).

The world is truth as the horizon of manifestation. However, it is a horizon *extrinsic* to the essence of manifestation, the ek-static "outside" within which everything appears, indifferently. It is within this indifference that the doubling of truth and the true occurs: "In the world everything and anything shows itself—children's faces, clouds, circles—in such a way that what shows itself is never explained by the mode of revealing specific to the world . . . . What is true in the world's truth in no way depends on this truth: it is not supported by it, guarded by it, loved by it, saved by it. The world's truth—that is to say, the world itself, never contains the justification for or the reason behind what it allows to show itself in that truth and thus allows "to be"—inasmuch as to be is to be shown" (Henry 2002, p. 16).

The truth of the world means, therefore, that truth is both the world manifesting itself and that which becomes manifest, made true by the horizon that makes possible its visibility. The true is *simply* what is brought to manifestation in the ek-stasis of the world, in the world as outside. Henry does not contest the world, but rather says of it that its way of phenomenalizing, which makes possible the manifesting of things simply because it acts as the outside in which everything appears *indifferently*, is not the only kind of phenomenality that is capable of *truth*. *If* truth is simple self-showing, and *if* the world is the horizon that makes possible all self-showing, then the truth of the world lies in this pure and simple coming-outside-of-itself to appear in the light of an indifferent "outside". In order to confirm something as "true", it suffices simply to confirm its appearing in the light of the world. To this way of construing truth Henry opposes what we can call "living the truth" or the truth of Life.

The truth of Life consists in Life's manifestation not in the light of the world but *in itself*, not in the *outside* but as *self-revelation*. Life is the phenomenological essence that confirms itself in its own living; it does not manifest itself but "lives itself", it generates and reveals itself. It is originary revelation. There is no cleavage between truth and the truth here, because the truth of Life is not an "outside" indifferent to what appears (thus, it is not true simply because it appears) but it is self-manifestation in experience, in its own *épreuve*. The suspicion that Henry is being tautological here is only illusory because "when it concerns the essence of Life, [self-revelation] means, on the one hand, that it is Life that

achieves the revelation, that reveals—but, on the other hand, that *what Life reveals is itself*" (Henry 2002, p. 29, author's emphasis).

*Truth of* the world, *truth of* Life. The possessive is decisive in both kinds of truth, producing in the former a separation between truth and the true and grasping in the latter the living heart of truth itself. However, it is one and the same way of intending the truth. Truth is the essence of self-manifestation insofar as such manifestation is possible starting only from itself. The truth of the world is therefore truth in an improper sense since it only ensures itself and the appearing of the phenomenon without saying anything about the essence of manifestation as such. By contrast, Life never appears as phenomena do, but rather self-attests; it feels itself and gives itself in such self-feeling and self-experiencing. In Life, truth and the essence of manifestation coincide, and in this way, there is no *indifference* between what appears and its self-manifesting. While the world is other than the phenomena that appear within it and that draw from it the possibility of their very appearing, Life is self-revelation. In actuality, it seems that we should only speak of truth in the context of Life, for if Life is manifestation grasped in its phenomenological purity, the pure essence of manifestation, then only in Life can truth and the true coincide: "What, then, is a truth that differs in no way from what is true? If truth is manifestation grasped in its phenomenological purity—phenomenality and not the phenomenon—then what is phenomenalized is phenomenality itself [ . . . ]. What manifests itself is manifestation itself" (Henry 2002, p. 25).

The world makes possible manifestation as its horizon but not as its *essence* because "the phenomenalization of phenomenality is a pure phenomenological matter, a substance whose whole essence is to appear—phenomenality in its actualization and in its pure phenomenological effectivity"(Henry 2002, p. 25). In this sense, there is truth in the world only in a secondary and improper sense, one derived from the authentic sense of truth understood as the essence of manifestation. For this reason, the world proceeds by *separating* truth and the true while Life *unites* them by its self-revelation. The world is not, in itself, the essence of manifestation but rather a horizon, a background that makes revelation possible but which does not reveal itself as Life can.

Once again, then, truth of the world, truth of Life: how should these two truths be understood? Is it a matter of a double truth after the model of parallel lines or of two paths born of the same origin? Henry writes that Life takes place and gives itself in the flesh, which is not a visible body that appears in the world but which is only felt. I, for instance, see and am seen as a body but I feel pleasure, pain, and so on thanks to the flesh that is not seen. A corpse, on the other hand, might have the same characteristics of a living body while not "feeling" anything. Life, therefore, manifests in the flesh that is felt as living, and the First Living is the Christ (see Henry 2002, p. 79), in whom the Life of every living is enfleshed. However, why are body/flesh and world/Life not simply parallel and untouching manifestations, but rather the manifestation of Life that takes place through *one and the same* living? What saves us here from the paradox of a duality that resides in one and the same living, as if the same body were an object on the one hand, and subjective Life on the other? To answer, it is perhaps necessary to go beyond Henry's inconspicuous Life and flesh, and back to what comes before any distinction and division, as Derrida posited Heidegger's *Frühe*/spark as coming before the distinction between *pneuma* and *spiritus*. While it may be true that Henry lacks a reflection on Spirit, overcoming Henry and returning to Spirit would allow us to think adequately the origin of different manifestations, since Life manifests itself in flesh, enlivening it and "giving it spirit". However, there is more.

If we return to Spirit, we might also be able to overcome the opposition between living flesh and the body that appears objectively in the world, for Spirit would be the origin of the two different manifestations of the same body. This would also explain why the objective body is worthy of respect both when living and when dead. Spirit is the origin which, as such, extends into all that it originates, even materiality, and renders the latter worthy of respect regardless of whether it manifests as objective body or as nature more

generally. There is no world that is merely objective and material on the one hand, and Life that is worthy of respect on the other, but everything is expression of Spirit insofar as it proceeds from the same *spiritual* origin. In this sense, we can read the Johannine Prologue along with Henry, who comments it in various places, and say of the "Logos Egeneito": "The Word was made flesh through the Spirit".

Spirit, then, is not what is opposed to matter but what makes possible both the materiality of the world and the inconspicuousness of Life. It makes both of these possible if we intend it as their origin by going beyond their opposition. In this way, Life can be Spirit—not in a Hegelian sense that would bind it to the Idea, but in the sense of what enlivens, as a spark that lights a fire that *moves* and makes everything be what it is. Just as Derrida writes that it is spark and origin that precedes the distinction between *pneuma* and *spiritus*, Spirit is at the origin of both conspicuous and inconspicuous manifestation by preceding all materiality and "spirituality". Spirit is such an origin because it makes things to be born and to take place, thus preceding all subsequent distinctions. As such, Spirit appears not as "something" but by means of its acting and operating, and thus Spirit *is* if and insofar as it operates. Now, to operate is to act, and acting is *praxis*. Therefore, *praxis* is not merely human action but is human action *as* the acting of Spirit. This is confirmed in the chapter of *I am the Truth* dedicated to the "second birth".

Since every living being lives because of Life's doing, Henry writes, "the relation of a living to Life cannot be broken, and cannot be undone" (Henry 2002, p. 162). That it cannot be undone implies that the event of Life must be incessantly possible and does not represent merely the moment of biological birth. In fact, a second birth is possible that "only comes about due to a mutation within life itself" (Henry 2002, p. 164), a self-transformation willed by the single life that leads it to its true essence, to absolute Life.[7] Henry continues: "That this transformation of life, owing nothing to world's truth or its logos, receives its motivation from this life, that it belongs to this movement and concretely accomplishes it, determines life as an *action*. *The self-transformation of life that it wills, consisting of an action and leading it back to its true essence, is the Christian ethic* [ . . . ]. [This] ethic presents itself from the start as a displacement from the realm of the word, meaning also of thought and knowledge, to the realm of action" (Henry 2002, p. 165).

The action of *praxis*, and of all doing, is founded in Life and in the hyper-power that it unleashes in its very acting. It is in works of mercy that the unleashing of the hyper-power of Life in action takes place, since these works are possible only by the event of "something other" in the life of each individual living. This is the sense of the Christian ethic and of the *praxis* that it introduces. It shifts the emphasis from the knowledge of God to the life and action that take place because of the Life of God. This is a decisive dislocation because, firstly, the truth of the world is no longer the truth of action. Secondly, in abandoning the paradigm of representative truth (and thus also the Hegelian paradigm), Christianity "unequivocally relates Life's truth to the process of its self-engendering, to the power of action. Finally, life is no longer about the power, the "I can" of the single ego, because all the living can act through "the process of Absolute Life's self-engendering" (Henry 2002, p. 166), which means doing the will of the Father. As such, "the ethic [links] the two lives, the ego's and God's, in such a way that it assures the former's salvation in practice. To do the Father's will designates the mode of life in which the Self's life takes place, so that what is henceforth accomplished in it is Absolute Life in its essence and by its requirements" (Henry 2002, p. 166).

"God is Life" and "access to God is access to Life itself, its self-revelation" (Henry 2002, p. 166). These are statements of Henry's that recur often. To clarify the essence of action, he writes that since action "consists in the application of a power", and since such power is only exercised thanks to Life, then all power is given to itself through Life (see Henry 2002, pp. 167–68). Salvation is not other than this access to Life, and "doing carries life as its irresistible presupposition, because there is no doing unless given to itself in life's self-givenness, unless the work of salvation is entrusted to it" (Henry 2002, p. 167). Not

just any kind of doing, that is, but a doing that leads to Life, not to the death that derives from all doing founded on the ego and on oneself.

With the distinction between these two kinds of action, Henry establishes the birth of a *praxis* that acts as mercy and as response to the kind of *praxis* which, by contrast, is the action that pertains to death and self-enclosedness. Unlike this second kind of *praxis*, which focuses only on the power that the ego attributes to itself, Christian action exemplified in the works of mercy is characterized by the forgetting of self. Thus, to the question of what kind of action characterizes the works of mercy, Henry answers that it is "the ego-defining power of the 'I can.'"(Henry 2002, p. 167). He writes that in this Ego "there is no power different from its own . . . except for the hyper-power of Absolute Life that gave [the ego] to itself by giving itself to itself."*In works of mercy—and this is why they are 'works'—a decisive transmutation takes place by which the ego's power is extended to the hyper-power of Absolute Life in which it is given to itself.*" (Henry 2002, p. 169, author's emphasis). In the final pages of this chapter, Henry specifies this sense of action, because in the Self, Life is in action. The true Self is not the finite self but is rather discovered in the forgetfulness of self in which alone Life's Self takes place. Simply put, in the forgetfulness of self that removes it from its being as a worldly ego, the living discovers its Self through the Life that enlivens it. The fulfillment of this forgetfulness is the "transmutation of the I" that only the works of mercy, the work in which it acts thanks to another, can bring about. "*Only the work of mercy practices the forgetting of self in which all interest for the Self (right down to the idea of what we call a self or a me) is now removed, no obstacle is now posed to the unfurling of Life in this Self extended to its original essence*"(Henry 2002, p. 170, author's emphasis). The exceptionality of action carried out as works of mercy is possible only because an Other (God as Life) is operative in our action, thus rendering it forgetful of self and of the belief that it owes its life only to itself: "Whether it involves nourishing those who are hungry, clothing those who are naked, caring for the sick, or another act, *the manner of acting in these various actions has ceased to concern the ego that acts or to relate to it in any fashion; a common trait equally determines them all: forgetting oneself*" (Henry 2002, p. 169). The refusal of this forgetfulness encloses the ego within itself and cuts it off from the "I can" that is the origin of the givenness of each individual life.

The source of *praxis* is thus Life, which leads the individual living to salvation through the forgetfulness of itself and through the rekindling of originary Life in its life. However, what brings the self to this forgetfulness? Why should the individual living turn to something other than itself? Why should it choose a different *praxis* that would bring it to salvation? Henry does not give answers to these questions, but nothing prohibits us from retaining the phenomenological model that he proposes (i.e., that each living is such because of Life) without opposing to one another two modes of manifestation (individual living and Life, world and Life, living flesh and objective body) and leading them back to their common root of Spirit as the Beginning-spark of all action.

In fact, the self is not a substance nor a transcendental Ego, but rather the originary acting that Spirit sparks into motion, thus making the self the "I can" of action and self-transformation. In short, Spirit makes possible the self-movement and self-transforming of the I. If the I moves toward Life, then just as we said that the Word is made flesh thanks to Spirit, we can now say that the ego is such through the Spirit that animates its action. This movement back to Spirit as the origin of the self (which, again, is absent in Henry) finally leads us beyond the Phenomenology of the Inconspicuous. This is because it shifts our thinking *away* from the inconspicuous *toward* which we are directed (Being, according to Heidegger) and *toward* what directs us there, i.e., the Spirit that constitutes the living self. In this sense, Spirit appears as the "soul" of *praxis* and action, Spirit that "is" insofar as it transforms, changes, moves, and animates the living and its *praxis*—the heart of the ego.

### 5.2. Spirit and Appel (Marion, Chrétien)

There is another phenomenon that moves beyond the phenomenology of the inconspicuous and beyond another opposition inherited from Jean-Luc Marion and Jean-Louis Chrétien. It is the double phenomenon of "call and response".

Marion writes about *l'appel*, the call or claim, already in 1989 in *Reduction and Givenness*, where he proposes it as a paradigm of phenomenological reduction against Husserl's transcendental reduction and Heidegger's existential one.[8] While Marion borrows the idea of the call from Heidegger, who writes of the "call of Being" (*Anspruch des Seins*), he thinks of the reduction itself as a call because the phenomenon appears for me as making a claim on me, as calling me (see Marion 1998, pp. 197–98) Call and response, or better, *appel* and *répons*,[9] become central themes in *Being Given*, especially in the well-known analysis of Caravaggio's work *The Calling of Saint Matthew*. In this painting, we do not witness Christ's call, but we understand from Matthew's astonished and inquiring expression that there has been a call and that Matthew is responding to it. What is more, his expression is what allows us to experience the call itself. The invisibility of the *appel* manifests phenomenologically in the response: "The a priori call awaits the a posteriori of the response in order to begin to have been said and to phenomenalize itself. The response states what the call had continually recalled to it" (Marion 2002, p. 287). The response is not only the condition of visibility, but it *makes visible* the call. It is not simply the response to a call, but *the very form of the call*, which appears as the response which makes the call visible in the first place, just as the responsorial belongs to the form of the psalm: "The responsal begins as soon as it has made the call phenomenalized. The call begins to appear as soon as it finds an ear in which to settle, as soon as the first 'Here I am!'" (Marion 2002, p. 288).

As we can see from these brief descriptions, the call is not a question that we simply hear and understand. Rather, it is only understood in the response. In and of itself, the call is inconspicuous, while what appears is the response. Nevertheless, we can pose the same challenge to this duality that we posed to Henry's oppositions of Life and world, Life and individual living, and so on: what keeps together the terms of these oppositions? From where do these terms originate? Before giving an answer, it is necessary to lead the call back to its originary pronouncement and, in this way, to open a dialogue with Chrétien, for whom call and response take the form of the *convocation* or *call to be*.

To return to the originary pronouncement, Chrétien reaches back to the Greek term for "call" or "claim", namely *kalein* (to call or call together), which Plato's *Cratylus* joins together with *kalon*, the beauty (see Plato 1997, pp. 101–56). *Kalein*, similar to the French *appeler* and the Italian *chiamare*, means both to give a name and to call upon. The call, which is undefinable because it does not request anything specific and does not ask for anything, nevertheless manifests itself, can appear in the *beauty* to which it gives its name (*kalein/kalos*).[10] "What is beautiful is what calls out by manifesting itself and manifests itself by calling out. To draw us to itself as such, to put us in motion toward it, to move us, to come and find us where we are so that we will seek it-such is beauty's call and such is our vocation" (Chrétien 2004, p. 9).

Beyond this etymological assonance, the call reveals the deepest sense of beauty, allowing the latter to be not seen, but heard. A phrase of Paul Claudel rings insistently like a refrain—"the eyes listen"—and conveys well the sense of beauty. " . . . Beauty is the very voice of things" (Chrétien 2004, p. 35), the "visible voice" (Chrétien 2004, p. 35) through which things "invite us to interrogate them" (Chrétien 2004, p. 36). The eye listens to the reality that calls upon it: an intertwining of vision and hearing in which the way we encounter reality finds its voice. The beauty that calls upon, that makes an appeal, is not a beautiful *thing*, but rather "beauty" is the name that Chrétien gives to reality's capacity to call upon us. It is due to this capacity that the method of knowledge by which we know is first of all the encounter with reality and with things. We are attracted to know things by the things themselves. The relationship with reality is therefore not a relation to objects, but a bond to which we are called. Reality calls upon us; this is beauty.

In order to clarify further the way in which he understands beauty, Chrétien cites a passage from the *Sermons on the Song of Songs* by Bernard of Clairvaux, where the beauty of Christ is invisible to the eye and revealed by faith: "The words he speaks are 'spirit and life'; the form we see is mortal, subject to death. We see one thing and we believe another. Our senses tell us he is black, our faith declares him fair and beautiful [ . . . ]. Black in the opinion of Herod, beautiful in the testimony of the penitent thief, in the faith of the centurion" (Chrétien 2013, p. 105). Beauty is reality according to its true sense and is not revealed only to the exceptional or to mystics. The sense and the truth of things becomes manifest through their call to listen, asking us to go beyond our vision and toward an infinite relation of hearing, to experience them as exceeding our sight. The real lets itself be known by attracting and calling upon us, according to its own truth, but—and here we find the antiphonary form of this relation—this call can be recognized and welcomed as such only "in our unavoidably belated response" (Chrétien 2004, p. 44).

The response is the way in which this silent call is revealed as such, as a call that calls upon. This call does not ask anything; it is a pure and simple call to the true sense of things. I can recognize beauty as a call "only if I respond in fear, in admiration, in bewilderment"; I recognize the call of the *Logos* "when I respond to it with thoughts and words" (Chrétien 2007, p. 15). Thought, or philosophy, is the response in which the truth that calls upon us reveals itself. However, what is it that appears when we answer to the call? In response to this question, Chrétien cites Saint Paul's *Letter to the Romans*: "Abraham . . . is the father of all of us . . . in the presence of the God in whom he believed, who gives life to the dead and *calls* into existence the things that do not exist".[11]

Chrétien comments on these verses extensively, as he finds them to be the clearest expression of how things come to be what they are. Everything, including the human itself, is called upon to be, and to exist is to respond to this call that reveals and manifests itself *in the response that existing itself is*. The beauty that calls upon is first and foremost God, who is beautiful because he calls things into existence. Following Pseudo-Dionysus on the common root of "to call" and "beauty", Chrétien comments that "the God of [Dionysus'] meditation and invocation is . . . superessential beauty, *huperousion kalon,* beyond being, who "in the manner of light, causes the beauty-producing communications of his initial ray to shine in all things. He calls *(kaloun)* all things to himself, and this is why he is called *kallos,* beauty" (Chrétien 2004, p. 15). A beauty which, in God, *is converted* into Goodness: "Springing into being, we answer . . . . It is no longer only the beautiful, but also now the good that *originates* in a call" (Chrétien 2004, p. 17, author's emphasis).

Thus, God "called all things from non-being into being when he spoke and they were made" (Chrétien 2004, p. 17). We can come thus far with Chrétien, who maintains the duality of the inconspicuous call and the conspicuous response. However, *how* does God call? To this, Chrétien does not give an answer, but nothing keeps us from answering that God calls as Spirit. In fact, there are not many alternatives to this answer: either the call takes place as a "voice" that we hear, or as an urge and solicitation to act. The latter fits the model that Chrétien introduces, since for him the call leads us to be, to be born, to live. To respond to the call is to live.

For Marion as for Chrétien, the response reveals that the call took place. Yet before the response, the call gives itself *as* a call to movement and to action. Before the response, the call gives itself and nothing prohibits us from understanding its origin as Spirit taking the form of a "call to Life". In this way, Spirit has its place at the origin prior to the distinction between call and response (or responsal), taking place as the origin from which the two terms must separate themselves in order to come to manifestation.

Once again, Spirit takes place as the possibility of Life, be this the Life described by Henry as the possibility of all living or the Life that is called to being by an inconspicuous origin. Spirit is the breath that precedes all distinctions and duality, the origin that gives origin to them.

## 6. Conclusions. Beyond the Phenomenology of the Inconspicuous: The Sense of a Path

These pages began with two questions that guided our path of thought: (1) How is Spirit manifest? (2) Does its manifestation as *not* an object, thing, or being connect Spirit to Heidegger's phenomenology of the inconspicuous? While the answer to the first question required the rest of the reflections above, the answer to the second was found to be negative almost immediately because the inconspicuous is not Spirit. From here, a tangential inquiry compared Heidegger and Hegel not on the inconspicuous, but on Spirit, and here Derrida offered a decisive help: his reading of Heidegger's commentary on Trakl's last poem showed that Heideggerian Spirit is *Frühe*, the initial spark that precedes all distinction between *pneuma* and *spiritus*, the animating spark that stands at the origin. Is it possible to develop a non-Hegelian phenomenology of Spirit understood in this way? This question required a long detour because, after Hegel, Spirit has not occupied the central position in phenomenology. At the same time, Hegel furnished an essential insight, namely that Spirit is Life.

This insight, along with the Derridian stress on Spirit as originary spark that precedes all distinction, guided our re-reading of some paths of thinking in phenomenologists who are considered heirs of Heidegger's Phenomenology of the Inconspicuous. These paths were, on one hand, the Life that is at the heart of Michel Henry's work, and, on the other, the call and response on which Jean-Luc Marion and Jean-Louis Chrétien have written. An explicit reflection on Spirit is absent in these French thinkers but, as I tried to show, Spirit can be understood as what *precedes* and *gives origin* to all subsequent dualities that are produced by the movement of manifestation. In the works we took up, these are the dualities of world-Life and call-response. In both cases, Spirit was posited as the *spark that gives origin to what subsequently differentiates itself by becoming manifest*. Now, where does this path lead us once we move beyond Hegel's phenomenology of (inconspicuous) Spirit and Heidegger's phenomenology of the inconspicuous?

Paradoxically, once Heidegger's phenomenology is left behind because it does not speak of Spirit, a new, non-Hegelian phenomenology of Spirit can finally appear. Here, Spirit is not what develops and unfolds, as in Hegel, but rather what *precedes*—not as the beginning point that is represented and thought through but as the spark and fire that gives origin and engenders Life. In Henry, Spirit is what animates Life itself, allowing the latter *subsequently* to separate into different manifestations, while in Marion and Chrétien, Spirit is manifest in the response as the origin that calls to being. Beyond the Phenomenology of the Inconspicuous is therefore what lies on *this* side of all appearing, namely, that Spirit that *appears* to be missing but is the core of Life, that which precedes the distinction between the material and the spiritual. In fact, Spirit precedes distinction as such because all distinctions derive from it. It is the living breath that calls to Life and thanks to which we are spiritual insofar as we live and begin to live again. From beginning to beginning, according to beginnings that will never end.

**Funding:** This research received no external funding.

**Institutional Review Board Statement:** I would like to use the Open Review option.

**Informed Consent Statement:** Not applicable.

**Data Availability Statement:** Not applicable.

**Conflicts of Interest:** The author declares no conflict of interest.

## Notes

<sup>1</sup> Heidegger (2003, pp. 134–36). Translator's note: the Italian translation of Heidegger's Zähringen seminar translates the German "anwest nämlich Anwesen" as "è presente infatti l'essere-presente", which itself translates in English to something like "present is namely Being-present", while the Anglophone translator opted to translate the first and third words in verbal form—not "present" and "Being-present", but "presences" and "presencing". It is due to this difference in translation that the author seemingly jumps from presencing to Being, but in fact the Italian text translates "Anwesen" as *Being*-present, and this explains why the author moves between presencing and Being so seamlessly in these pages on the Zähringen seminar.

<sup>2</sup> See Hegel (2010), *Doctrine of Notion*, Section Three: *The Idea*, chp. 1: *The Life*.

<sup>3</sup> Husserl (2001): "Die philosophischen Probleme von Natur und Geist stehen nicht etwa getrennt neben den naturwissenschaftlichen und geisteswissenschaftlichen". p. 15.

<sup>4</sup> Derrida (1989, p. 113). It should be noted that, according to Derrida, Spirit is not the original and nothing precedes the *différance*. In these pages we are proposing the comment that the French philosopher makes about Heidegger and the Spirit as it is introduced in Trakl's poem by the German philosopher. Derrida is perhaps the one who, better than others, has grasped this aspect of Heidegger's work.

<sup>5</sup> See "Independence and Dependence of Self-Consciousness: Lordship and Bondage" in Hegel (1977).

<sup>6</sup> See Book VI of Aristotle's *Nicomachean Ethics*.

<sup>7</sup> Translator's note: Henry distinguishes between "Life" and "life". The former is what self-manifests immanently in the flesh, what makes the difference between a living human being and a corpse. The latter is what we commonly understand as life, the life of the individual ego. Because Life is the truth of all phenomenological manifestation, and because Life manifests as the self-feeling of living flesh, life as belonging to an individual ego is an illusion that results from the ego's forgetfulness of the Life that precedes it and gives the ego to itself. The English translation of *I am the Truth* marks this distinction through capitalization—"Life" as opposed to "life".

<sup>8</sup> Translator's note: the English translation of *Reduction and Givenness* translates *appel* both as "claim" and as "call", depending on which term is more appropriate in the given individual case. In the French, Marion plays on this ambiguity quite a lot, and this play on the double meaning of *appel* is somewhat lost in English.

<sup>9</sup> Translator's note: "*Répons*" is a neologism and technical term for the response that makes visible the call and, in this way, is the "very form of the call". The ordinary French word for "response" is *réponse*. In the English translation, the neologism *répons* is rendered as "responsal" to highlight the difference between an ordinary response and a response that makes the call visible.

<sup>10</sup> And giving its name to beauty, "the call designates that which is essential to beauty, the very nature of its manifestation". Chrétien (2004, p. 9).

<sup>11</sup> *Romans* 4: 16–17 (NRSVCE), my emphasis. In Greek, "who calls" is *kalountos*, in Latin *qui vocat*.

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
