# Peer review of "Beyond the Phenomenology of the Inconspicuous†"

_religions, doi:10.3390/rel12080558_

Round 1

Reviewer 1 Report

The author makes a compelling argument for the inconspicuousness of the Spirit as the active source for being/body-soul.  However, the introduction needs to state and explain more clearly the agenda of the article ; that is, because the inconspicuousness of the Spirit, Hegel and Heidegger inadequately account for it and that Marion, Henry, and Chretien's Christian understanding of Spirit more adequately account for it.  

There are stylistic and textual issues to address: 1. lines 94-105 is a block quote; 2. lines 166-67 need a verb; 3. line 167 needs to specify what "this" refers to; 4. need to break up the long paragraph of lines 166-204;  5. line 179, specify the references to "vague allusions"; 6. on line 185, to what does "of course" refer?; 7. lines 198-204 seems to have an implicit contradiction; rethink the sentence; 8. lines 211-218 needs to be clearer; 9. lines 245-249 are awkwardly written; 10. need to break up the long paragraph of lines 335-359; 11. the rhetorical questions in lines 367-371 are unclear; they would be more clear if rewritten into indicative statements.  

Author Response

Dear colleague, 

Thank you so much for your comments. 

In order to better clarify the intent of the paper, I will explain my purpose more rigorously in the Introduction. 
Thanks again for your careful and valuable reading

Reviewer 2 Report

I think it can be published as is, but I have two suggestions, along with a couple of typos.

  1. It seems a little disingenuous to appeal to Derrida, particularly twice, for a defense of Spirit as origination. For Derrida, that which "precedes and originates all differentiation" is différance. His appeal to différance (and to Khora) are designed to work precisely against what you are doing. At least mention that the deep divergence between Spirit and Khora/ différance. Since they are so different, what is the point of the appeal to Derrida?
  2. Our preoccupation with the inapparent (inherited from Heidegger) functions to reinforce and hide the problems in the bifurcation between Call and Response in Chretien and Life and World in Henry (and Visible and Invisable in Marion). Thus, your appeal to Spirit is extremely valuable. But if we accept it, doesn't that also undermine the absolute indifference between horizon and thing (line 305). Why does Henry accept that "the horizon is extrinsic to the essence of manifestation, the ek-static 'outside' within which everything appears indifferently"?  Do you accept this? Later (378), you claim (rightfully I think) that your appeal to Spirit overcomes the absolute opposition between flesh and body (See also Falque, Wedding Feast of the Lamb, etc). Doesn't it also overcome the absolute opposition between horizon and thing (such that we no longer need to accept the violence of Heidegger's opposition between Earth and World)?
  3. Typos   

line 355: there (is) truth in

line 422: I can (comma)

line 447: "specify even better" is awkward.

line 454: (is) now removed (?)

line 479/80. "toward" repeated twice is awkward.

Author Response

Your remark about Derrida is quite right: I made my appeal to Derrida because he is the one who best of all (or the first, in my opinion) to have underlined that Heidegger speaks of the inapparent Spirit in “Unterwegs zur Sprache”. The appeal to Derrida is limited to the text in which he proposes his comment to Heidegger and I agree with your statement, that is: «"which" precedes and originates all differentiation "is différance"». In the paper, I will better explain the appeal to Derrida and what is suggested.

With regard to the important observation concerning Michel Henry and the "outside" of the world, I disagree with the Henry’s analysis of the world and I believe that we need to recover a way to think about exteriority. I agree with Emmanuel Falque and his remarks about the henryen separation between “chair” and “corps”. However, I think that to highlight  the theme of "spirit" is consistent with Henry's work because the French philosopher, precisely in the “Essence de la manifestation”, takes into account the Hegelian “Phenomenology of the Spirit” that he criticizes because the German philosopher fall within ontological monism. However, Henry recognized that Hegel, better than others, had grasped the impossibility of separating essence and manifestation. The Hegelian way is not followed by Henry, but it is by commenting on the Hegelian Spirit that Henry writes the counterpoint and becoming "Michel Henry".

Thank you so much for your suggestions and your valuable review